Distribution and biological implications of plastic pollution on the fringing reef of Mo’orea, French Polynesia

Connors Elizabeth J. bethconnors@berkeley.edu
Marine Science and Integrative Biology, University of California , Berkeley , CA , United States of America
Reimer James
Electronic publication date: 2017 Aug 29
Publication date: 2017
Volume: 5
Electronic Location ID: e3733
Received 2017 Jan 2; Accepted 2017 Aug 4
Copyright: ©2017 Connors
Copyright year: 2017
Copyright holder: Connors
License: This is an open access article distributed under the terms of the Creative Commons Attribution License, which permits unrestricted use, distribution, reproduction and adaptation in any medium and for any purpose provided that it is properly attributed. For attribution, the original author(s), title, publication source (PeerJ) and either DOI or URL of the article must be cited.
License URL: https://creativecommons.org/licenses/by/4.0/

Keywords: Plastic, Pollution, Coral, Microplastic, Corallimorph

Funding: The author received no funding for this work.

==============================
Coral reef ecosystems of the South Pacific are extremely vulnerable to plastic pollution from oceanic gyres and land-based sources. To describe the extent and impact of plastic pollution, the distribution of both macro- (>5 mm) and microplastic (plastic < 5 mm) of the fringing reef of an isolated South Pacific island, Mo’orea, French Polynesia was quantified. Macroplastic was found on every beach on the island that was surveyed. The distribution of this plastic was categorized by site type and by the presence of Turbinaria ornata, a common macroalgae on Mo’orea. Microplastics were discovered in the water column of the fringing reef of the island, at a concentration of 0.74 pieces m−2. Additionally, this study reports for the first time the ingestion of microplastic by the corallimorpha Discosoma nummiforme. Microplastics were made available to corallimorph polyps in a laboratory setting over the course of 108 h. Positively and negatively buoyant microplastics were ingested, and a microplastic particle that was not experimentally introduced was also discovered in the stomach cavity of one organism. This study indicates that plastic pollution has the potential to negatively impact coral reef ecosystems of the South Pacific, and warrants further study to explore the broader potential impacts of plastic pollution on coral reef ecosystems.

Introduction

Anthropogenic debris is accumulating at a rapid rate in coastal and oceanic ecosystems worldwide (Critchell & Lambrechts, 2016). The most significant contributor to these growing marine litter deposits is plastic (Islam & Tanaka, 2004), a material widely used by humans (Eriksen et al., 2013). In the marine environment, plastic is initially buoyant, and easily dispersed over long distances via wave action and wind (Derriak, 2002). Floating plastic is detrimental to both ecosystem nutrient cycling and marine wildlife, as it can absorb and secrete chemicals (Moore et al., 2001; Islam & Tanaka, 2004). Larger plastic pieces (macroplastic > 5 mm) are degraded by UV light and wave action into microplastics (<5 mm) (Gregory, 1999; Eriksen et al., 2013). Plastic pollution, especially microplastic, is often confused for food by marine organisms (Avio, Gorbi & Regoli, 2016), and absorbed toxins in plastic pollution can bioaccumulate in higher trophic levels (Rochman et al., 2013; Farrel & Nelson, 2013). High concentrations of floating plastic debris have been reported in many areas of the ocean (Cózar et al., 2014; Eriksen et al., 2014), in particular at the center of oceanic gyres (i.e., the great “garbage patches” Berloff & McWilliams, 2002). The South Pacific gyre is dominated by microplastic particles (plastic < 5 mm in diameter) (Eriksen et al., 2013).

The islands of the South Pacific are sinks and sources for plastic in the marine ecosystem. Uninhabited islands in the area accumulate plastic debris at alarming rates, acting as sinks for plastic from the South Pacific gyre (Lavers & Bond, 2017). Most of the inhabited islands in this region are also a source of plastic entering the marine environment, as their landfills are generally uncontrolled tipping locations on or near the coast (Morrison & Munro, 1999; Hayes & Richards, 2010). In French Polynesia, which has the most technologically advanced waste management center of the whole Pacific region, it is unclear if the amount of plastic pollution entering the marine environment has increased as funding for their landfill has shifted from the French government to local municipalities since 2010 (Morrison & Munro, 1999; Hayes & Richards, 2010).

Coral reefs, a common coastal ecosystem of South Pacific Islands, are productive and biologically diverse (Mumby et al., 2016; Trapon, Pratchett & Penin, 2010). Although coral reefs cover 0.2% of the ocean floor, they contain around one third of all described marine species (Reaka-Kudla, 1997), and millions of people in the South Pacific depend on coral reefs for food (Costanza et al., 1997). Coral reefs are threatened by anthropological effects on a local and global scale, including sedimentation, increasing temperatures from global climate change, and changes in sea water chemistry (Wilkinson, 1999; Fitt et al., 2001; Hoegh-Guldberg et al., 2007; Anthony, 2016). Macroplastics such as fishing nets are known source of coral degradation (Donohue et al., 2001), and microplastics contaminate the reefs around Australia, including the Great Barrier Reef (Reisser et al., 2013; Hall et al., 2014). It is unclear how the magnitude of plastic pollution on the fringing reef of a South Pacific Island would compare to Australian coastal waters.

Our knowledge of the biological impact of plastic pollution on coral is limited. The ingestion of microplastic by scleractinian (reef-building) corals has been observed in a laboratory setting (Hall et al., 2014), but never in situ. The ingestion of plastic by a large-bodied, non-calcifying coral (Lin et al., 2016; Kuguru et al., 2007), Discosoma nummiforme was examined in this study. Corallimorphians including D. nummiforme are more resilient to temperature and rising CO2 levels than scleractinian corals (Medina, 2006; Kuguru et al., 2007; Norstrom et al., 2009; Veron et al., 2009), and may ingest plastic particles at a slower rate than scleractinian corals.

Categorizing the distribution of plastic pollution on the South Pacific island of Mo’orea, French Polynesia, as well as understanding the interaction between plastic pollution and resilient corals like corallimorphs will help us understand the magnitude of the threat that plastic pollution poses to coral reef ecosystems in the South Pacific region. This work specifically aimed to (1) understand the extent of the current macroplastic pollution problem on the South Pacific island of Mo’orea, French Polynesia (2) determine the concentration of microplastic particles in the water column of the fringing reef of Mo’orea and (3) evaluate if plastic particles are ingested by the corallimorph Discosoma nummiforme.

Figure 1 Location and study sites of Mo’orea Island.

Methods

Island-wide field survey of macroplastic

Mo’orea (17 30′S, 149 50′W), French Polynesia, has a fringing coral reef that encircles the whole island (Fig. 1). The three main populated centers on the island are Afareaitu (2012 census pop. 3,455), Haapiti (pop. 4,062) and Pao Pao (population 4,580) (Brinkhoff, 2012; see Fig. 1). Plastic surveys were conducted over the course of six weeks, from October 10 to November 20 2016, on beaches around Mo’orea (Fig. 1). Additional plastic sampling was conducted at the two largest public beaches on the island: Plage Public de Ta’ahiamanu (17 29′32.7″S 149 51′00.8″W), and Temae Beach (17 29′54.5′S, 149 45′42.9′W), The laboratory study was also conducted on the island, at the Richard B. Gump Research Station (17 29′25.1″S 149 49′35.5″W, Fig. 1).

Macroplastic (plastic > 5 mm) abundance was surveyed on Mo’orea’s perimeter road kilometer markers (called PK markers) (Fig. 1). On the northern side of the island, the waterfront (typically beaches or river outlets) at every PK marker of the perimeter road were surveyed, and around the remainder of the island, the beaches were surveyed at every 3rd PK marker, due to accessibility issues (the waterfront was not accessible from the road at every PK marker, especially on the southern side and northeastern corner of the island). The PK markers were used to randomize the types and location of sites categorized. Upon reaching a site at the PK marker, the site was first categorized by anthropogenic “site type”: residential, hotel, natural or public beach site (Of the 26 sites visited, ten were classified as residential sites, four as hotels, four as public beaches and eight as natural sites, see Fig. 2). The presence or absence of the abundant macro algae Turbinaria ornata was then recoded. This macroalgae was recorded in the study because pilot data indicated that this algae forms mats on the surface of the water, in which plastic is often entangled. The residential and natural beaches were categorized as either non-T. ornata or T. ornata beaches, but the five river outlet sites and the four hotel sites surveyed were counted as separate categories in this analysis, as water movement in river outlets and hotel clean-up efforts affected T. ornata presence at these sites. Once the beach was categorized by anthropogenic site type and presence/absence of T. ornata, a timed five-minute trash pickup was conducted. Two researchers, beginning at exactly the PK marker, walked along the beach in opposite directions, collecting all plastic pieces within 5 m of shore, for the survey. At each site, after the number of plastic particles collected on the beach in the interval was recorded, the percent of plastic of the total pollution found at the site was estimated. Percent plastic of all pollution and total number of plastic pieces collected in the surveys were quantified separately because they were not correlated (linear regression p > 0.5). Additionally, the distance in km between the nearest population center (Afareaitu, Haapiti, or Pao Pao, depending on which was closest to the site) and the beach site was recorded.

Figure 2 Number of macroplastic pieces collected in five-minute surveys, according to (A) site type and (B) the presence of T. oranata.

Microplastic survey

A plankton net (mouth size 0.07 m2; mesh size 0.05 mm) was used to collect water samples at Plage Publique de Ta’ahiamanu, to test for the presence of microplastic in the water column. A total of six 3-m plankton tows were conducted at the surface of the water in the intertidal zone of the beach, at randomly chosen intervals (of approximately 5 m) parallel to shore. The total area of water surveyed was calculated using the formula: A = l∗w, where l is the diameter of the net (0.3 m) and h is 18 m (3 m × 6 trials). The area of the water surface, as opposed to a volume of ocean water, was calculated to homogenize the results with similar studies on coastal plastic. Microplastics were identified under a light microscope at the Gump Research Station. Each piece discovered was measured and photographed, and the number of plastics was divided by the surveyed area to determine microplastic concentration.

Corallimorph plastic ingestion experiment

The plastic used in the study was collected from Temae beach sand (Fig. 1) and isolated from the bath product “LAINO Exfoliating shower gel”. The naturally collected plastic found in the sand was used in the study because of its negative buoyancy; 5 mm pieces of plastic were taken to Gump Station, and smashed with a hammer until they matched the size of observable microplastic in the water column (0.2–1 mm diameter). The polyethelene plastic beads in the shower gel were isolated by water filtration in a 0.005 mm plankton sieve. All of the isolated beads were a uniform size of 0.2 mm, green in color, and positively buoyant. Plastic color was used to differentiate the buoyancy of the plastic, as the positively buoyant plastic beads from the shower gel were green, while the negatively buoyant plastic collected at Temae were blue.

A total of 44 corallimorph polyps were collected from the fringing reef in a depth range of 1–3 m of water at Plage Publique de Ta’ahiamanu (Fig. 1) Corallimorphs were identified to a species level following the descriptions of Fautin and the Mo’orea Biocode database as Discosoma nummiforme (Paulay, 2007; Fautin, 2012). The disconnected polyps (average diameter 5 mm) remained attached to coral rubble rocks for the duration of the experiment. At the Gump Station, the polyps were given one week to adjust to laboratory conditions in a large flow tank (28 °C unfiltered ocean water), then thirty-four were placed in a separate experimental tank (28 cm *30 cm *8 cm) where they were exposed to plastic. Ten organisms were maintained as control.

In the experimental tank, 0.5 mL of both the lab-isolated and of the naturally collected plastic (1 mL total plastic) haphazardly placed on and around the corallimorph polyps. The water flow in the tank containing the corallimorphs was stopped during this procedure. Flow reduction of and the high plasitc concentration were used to ensure the most ideal conditions for plastic consumption. After 84 (n = 19) or 108 (n = 15) hours, the polyps were dissected, and number and color of plastic present in the organisms’ tissues were recorded.

Statistical analysis

Non-parametric tests were used as the number of samples per site type were not equal. Differences in amount of macroplastic present on beaches among the different beach classifications were examined for significance using a Kruskal–Wallace test, and a post-hoc Kruskal–Nemenyi test. To test for differences in macroplastic abundance in the presence of Turbinaria, a Kruskal–Wallace test was used. To test if distance from population centers on the island was predictive of macroplastic concentrations on sampled beaches a linear regression was used. For the microplastic feeding trial data, a Wilcoxon Rank Sum Test was used to examine the significance of the differences in natural-caught negatively buoyant plastic vs isolated, buoyant plastic over the time periods. This test was also used to examine the significance of the different hourly rates of consumption. All statistical tests were conducted in R (R Core Team, 2016).

Results

Island-wide field survey of macroplastic

Plastic was found on every beach surveyed on the island. Other than plastic, the most common pollutants were glass, metal and cloth. Public beaches, natural areas and residential areas had similar amounts of plastic (45 ± 5 SD). These three site types had ten times higher levels of average number of plastic pieces than hotel beaches (4.75 ± 1 SD). The differences of means in plastic amount collected varied by site type (Kruskal–Wallace, Chi-sq = 9.6, 3 df, p < 0.05, Fig. 2A). In the post-hoc analysis, the mean of plastic collection on hotel beaches was lower than natural and public sites (p ≤ 0.05), but the mean amount of plastic collected on hotel beaches was not significantly lower than residential beaches (Posthoc Kruskal–Nemenyi, p > 0.1). Of the 26 sites, 17 were natural beaches, of which 7 had Turbinaria, and 10 had no Turbinaria. 29% more plastic was present on beaches that had Turbinaria present than on clean beaches (Posthoc Kruskal–Nemenyi test, p < 0.05). When compared to the remaining sites (river outlets and hotel beaches), mean plastic amount was higher on beaches with Turbinaria present. (Kruskal–Wallace, Chi-sq = 15.0, 3 df, p < 0.01, Fig. 2B). The percentage of plastic on the beaches ranged from 20% to 100% of the total pollution present, with an average of 68% plastic pollution. The number of plastic pieces, as well as the percentage of plastic of all waste found on the beaches, increased with distance from population centers, but not significantly (linear regression for both, p > 0.05, Fig. 3).

Figure 3 Linear regression between distance from population centers (km) and both total plastic pieces collected in 5 min (grey triangles) and percentage of plastic of total trash found (black circles).

Microplastic concentration

Microplastic was found in the water column at the collection site, with a total of four pieces found in the six tows. One of the four small pieces of plastic was assumed to be, based on color and texture, from a larger piece of plastic collected in the tow. Overall, the six tows contained 0.74 pieces of microplastic m−2 surface area.

Exposing corallimorphs to microplastic

Of the 34 corallimorph polyps exposed to plastic, 19 (55%) polyps ingested one or more plastic particles during the experiment. The number of plastic particles ingested by individual polyps varied from zero to eight. The average amount of plastic consumed in the shorter time trial was 0.7 pieces/polyp and in the longer time trial 1.5 pieces/polyp. The mean amount of total plastic ingested did not vary with treatment time (Wilcoxon rank sum test, p > 0.05, Fig. 4). Although the total amount of plastic did not vary over the treatments, the amount of the positively buoyant plastic (the green micro-bead plastic) consumed increased over the separate time trials. In the first treatment time of 84 h, an average of 0.7 blue plastic particles (BPP)/polyp were consumed by the corallimorphs and 0 green plastic particles (GPP)/polyp were consumed. In the longer trial of 108 h, an average of 0.9 BPP/polyp and 0.5 GPP/polyp were consumed. The GPP consumed varied significantly over the separate time trials. (Wilcox rank sum test, p < 0.01, see Fig. 4.) Additionally, a yellow plastic particle was found in a polyp in the first treatment, presumably present in the tissue before the experiment (see Fig. 4). The ten control corallimorph polyps consumed zero plastic particles, experimentally introduced or otherwise.

Figure 4 Number of plastic particles consumed by Discosoma nummiforme according to treatment time and plastic color.

Discussion

Plastic accumulated most on natural beaches, and the amount of plastic and percentage of plastic debris increased insignificantly with distance from major population centers. This distribution of plastic on the island is somewhat surprising when compared to the literature, which generally find larger litter loads near urban areas (Garrity & Levings, 1993; Ryan et al., 2009). While natural areas had a higher than expected amount of plastic, the large volume of plastic on the public beaches of Mo’orea supports previous literature on coastal plastic debris. A recent review article on coastal pollution categorized over 60% of coastal pollution as from sources of “shoreline and recreational activities” that are prevalent on public beaches (Vennila, Jayasiri & Pandey, 2014). Finally, little scientific literature exists on hotel management of plastic debris, but the low amount of plastic discovered on hotel beaches is likely explained by hotel staff working to maintain a clean, white and sandy beach, as expected by hotel guests (D Venuit, pers. comm., 2016).

The beaches contaminated with Turbinaria ornata had a significantly higher amount of plastic than beaches without it. In the study, the plastic pieces collected were often caught in the thallus of this algae. The comingling of marine organisms and plastic is not a new phenomenon, as invertebrates and algae have been associated with plastic (Whitacre, 2012). Retention of plastic by T. ornata mats floating at the water’s surface may in fact account for the low presence of plastic found on beaches near population centers, as floating T. ornata mats can travel long distances (Martinez et al., 2007), and the currents of Mo’orea change rapidly (Hench, Leichter & Monismith, 2008). Further investigation into the association between plastic pollution and T. ornata is necessary to fully understand their association.

Surface waters of the intertidal zone of Mo’orea are contaminated with small plastics. This study adds to the evidence in the literature that suggests microplastics are the most abundant type of debris in all marine environments (UNEP, 2016). The concentrations of microplastic is much higher in oceanic gyres than it is on Mo’orea; as high as 334 pieces m−2 in the Northeast Pacific (Moore et al., 2001), and 396 pieces m−2 in the center of the South Pacific gyre (Eriksen et al., 2013). The concentration of plastic found in Mo’orea’s intertidal zone, however, is on the same order of magnitude as studies conducted outside of oceanic gyres of different geographic areas. For instance, the concentration in the Caribbean Sea (1.414 pieces m−2), and the concentration in the Gulf of Maine (1.534 pieces m−2) are both are slightly higher, but of a consistent magnitude when compared to the found concentration of 0.74 pieces m−2 in Mo’orea (Law et al., 2010). This equivalent magnitude of plastic on Mo’orea and the North America and Carribean coast is unexpected, as the concentration of plastic off the coast of a highly developed area should have a higher magnitude of plastic than an island in the South Pacific. Off the coast of Australia researchers found that plastic accumulates in waters near population centers (Reisser et al., 2013). Papeete, the capital of neighboring island Tahiti, is a relatively densely populated city, and likely has a large amount of plastic use, that may contribute to the plastic on Mo’orea. Any future study of the intertidal zone should include a larger portion of Mo’orea’s and even Tahiti’s coastline, covering km rather than meters of water, to further investigate microplastic accumulation in the region.

This study demonstrated, for the first time, the ingestion of microplastic by the corallimorphia D. nummiforme. The corallimorph polyps in the study ingested plastic particles at a slower rate than the scleractinian coral from Hall et al. (2014) paper (55% polyps inundated in 108 h, versus 21% polpys in 12 h). Although plastic consumption occurred at a slower rate, plastic presence in both studies caused considerable mucus formation that may represent an additional energy expense associated with microplastic contamination (pers. obs.; Hall et al., 2014). In both studies, plastic particles were found within the mesenterial tissue upon dissection, which may impede the digestion of these organisms (Hall et al., 2014).

In the experiment, the negatively buoyant plastic was ingested more often by these benthic organisms. The ingestion of the positively buoyant plastic by the corallimorph polyps is more surprising, especially because it only occurred in the longer treatment time (108 h). It has been previously hypothesized that corallimorphs prey on zooplankton or absorb dissolved organic material at an increased rate to survive temperature changes (Kuguru et al., 2007), so it is possible that under the induced stressful conditions (no water flow), the polyps altered their feeding rate and consumed more plastic. It is also possible that the green microbeads began to sink over time, and as they entered the benthic environment the polyps consumed them as apparent food. The high concentration of plastic and low flow in the experiment were ideal conditions for plastic consumption; future studies more similar to the actual reef environment are necessary for predicting rates of plastic consumption by corallimorphs in situ. These studies are essential especially because in this study, the presence of yellow plastic in the stomach cavity of D. nummiforme demonstrates that plastic is being ingested by corallimorphs on the reef of Mo’orea.

In conclusion, plastic pollution is prevalent on the beaches and reef of Mo’orea, as it was found on every beach visited during the study. Macroplastic was found in significant higher amounts on beaches with the algae T. ornata, and future study is warranted to understand if plastic pollution is transported around the island by this alga. Microplastic pollution was found in the water column of the fringing reef, and in the stomach tissue of the prevalent reef organism D. nummiforme. Under laboratory conditions, both buoyant and non-buoyant plastic were ingested by this corallimorphia. Further study into the distribution and biological consequences of plastic pollution on Mo’orea, and neighboring islands, is necessary to understand and combat this ongoing problem.

Supplemental Information

Supplemental Information 1 Raw code for figures

Click here for additional data file.

Data S1 Raw data

Click here for additional data file.

The author thanks Dr. Jonathon Stillman, Dr. Justin Brashares, Dr. Cindy Looy, Dr. Patrick O’Grady (UC Berkeley) and three anonymous reviewers for their valuable comments on the manuscript; Mr. Eric Armstrong, Mr. Ignacio Escalante Meza, Dr. Natalie Stauffer-Olsen and the Gump Station staff for their assistance with field logistics; and Mr. Eric Witte, Ms. Charlotte Runzel and Ms. Jacey Van Wert for the assistance sampling corallimorphs.

Additional Information and Declarations

Competing Interests

Author Contributions

Data Availability

The author declares that they have no competing interests.

Elizabeth J. Connors conceived and designed the experiments, performed the experiments, analyzed the data, contributed reagents/materials/analysis tools, wrote the paper, prepared figures and/or tables, reviewed drafts of the paper.

The following information was supplied regarding data availability:

The raw data has been supplied as a Supplementary File.

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
