# Peer review of "Distribution and biological implications of plastic pollution on the fringing reef of Mo’orea, French Polynesia"

_PeerJ, doi:10.7717/peerj.3733_

## Round 0.1 · original submission · Major Revisions

Two reviewers have gone over your submission, and while both had positive things to say about the work, they also offered many constructive critical comments. In particular, the English of the paper is far too colloquial/casual and is in need of extensive revision by a trained marine biologist; please ensure this is thoroughly and carefully undertaken. Additionally, I agree with the large majority of reviewer 2's numerous comments on your study.

Finally, I notice in several places you call Discosoma a "soft coral" - this is taxonomically incorrect, and in other places you call these "Discosoma spp." - implying multiple species. Please identify your animals to species level and provide information on how the animals were identified; otherwise the potential for conducting experiments with a mixed pool of species remains.

Thus, due to numerous comments from the reviewers and my own concerns, my decision is "major revisions" - this was almost a "reject" -but as your work is novel and under-reported I think it deserves a chance to be published if given the proper corrections. The revision will likely be long and hard work, but I wish you good luck and look forward to seeing a new version of your paper.

Reviewer 1 ·

Basic reporting

- Line 139 Sentence is confusing, better: “In fact, they evolved from hard corals during the Cretaceous period; a period typified by high CO2 levels.”

- Line 155: Please correct “frigning” to “fringing”

- Line 391 "The blue plastic", not "FTHe blue plastic"

- Figure 1: Please indicate direction in the figure itself, as you mention “northern side” (Line 172) in the text. In Line 206 it is mentioned that the Richard B Gump Research Station is shown on Figure 1, but it is not, please correct that. It might be useful to include longitude and latitude in Figure 1, as you state in the text for every place.

- Caption to Figure 1: Please be more specific in the caption of Figure 1,” Map of Mo’orea showing locations of macroplastic surveys (black stars), the collection site for blue plastic, and the collection site for corallimorphs and microplastic surveys” mentioning which stars correspond to the collection site for blue plastic, corallimorphs and microplastic survey, just like you do in the text and for the black stars in the caption. With these corrections the figure and caption can stand alone without needing the text for explanations.

Experimental design

- Line 171 Please indicate the size range of what you consider macroplastic.

- Line 210-213 If you say, a volume would have been better to represent the study, why did you not use a volume in addition to the surface? I understand that you want to homogenize your results with other results in the literature, but could you please connect the two sentences in a causal manner.

- Line 299-300 Since the yellow plastic was clearly ingested before the experiment, presumably it is not included in the Wilcoxon rank sum test, correct?

Validity of the findings

Line 313. Nice discussion of possible reasons for the weak correlation between pollution and closeness to populated areas (Figure 5 and 6). The explanation to why this weak correlation is contradictory to literature on plastic pollution is satisfactory. The conclusions make sense and are supported by given literature, even if not yet researched in this context of plastic pollution, which is discussed as well.

Additional comments

The study would have benefited from a more robust design, with a larger sample size, both in microplastic tow and corallimorphs collected. The macroplastics sampling could have been done over a longer period of time, as well as with in a clear beach transect (with a determined size).
But for the data collected and results presented the manuscript is clear in hypothesis, with the results and discussion connecting to them. It is well written in professional and clear English.
With the corrections above the paper can be published without further review.

Reviewer 2 ·

Basic reporting

Although there are no problems with English language in general, professional English is not used. A general review of the text is necessary to improve the general clarity of the manuscript.

References are used inadequately at several points of the texts.

Experimental design

This paper shows for the first time the ingestion of microplastic by soft coral Discosoma spp. The research question and experimental design are adequate, but the methods require more detail and information to assure replication.

Validity of the findings

The findings are valid, but the results, discussions and conclusions need to be better explored and more clearly stated.

Additional comments

This paper shows for the first time the ingestion of microplastic by soft coral Discosoma spp. The impact of plastics on marine organisms has been well documented for some animal groups, but had not yet been evaluated for soft corals. It is an interesting contribution to our understanding of the impacts of this type of pollution and the manuscript has potential, but needs to be substantially improved in order to be acceptable for publication.

Although there are no problems with English language in general, professional English is not used. The author tends to use overly wordy sentences, repetition, and clarity if often lost. A general review of the text is necessary. Remove unnecessary words to make the text clearer, and avoid repetition of the same words throughout or over consecutive sentences. Several sentences could be merged to avoid this.

Several modifications to the abstract and introduction are proposed below. The methods section is not very descriptive and needs to provide more details. Results and discussion need to be better explored.

Abstract
In general the abstract is clear, although some rewording is necessary. I also suggest focusing more on the obtained results.

Line 47: Not only of the open ocean. I suggest including coastlines as well.

Line 51: Change “During the course of this study” to the time interval of sampling efforts.

Line 55: Change to: “To test the impact of plastics collected from reef on corals, microplastics were made available to the soft coral Discosoma spp. in a laboratory setting.”

Line 56: a species of soft coral Discosoma spp.? Spp. Indicates more than one species…

Line 57: “The resilience of Discosoma spp. to fluctuating temperatures and rising…” Change this throughout the manuscript.

Lines 57 to 63: Avoid these many repetitions of “Discosoma spp.”

Line 60: “Positively and negatively buoyant…”

Line 62: “…microplastic that was not experimentally introduced was found in the stomach cavity of the organism.” Just one? Corals are colonies…

Line 64: And what about the ecosystem? Include a broader potential impact.

Introduction
Line 67 to 70: Change “deposits” to “accumulations”

Line 67 to 70: Avoid beginning two consecutive sentences with the same word. This happens throughout the paper.

Line 72: Again, avoid using the term “deposit”.

Line 73: “…accumulate on the substrate and act as a partition that inhibit benthic gas…”

Line 76: Change to “adsorb”, and modify “secrete”. Plastics are not organisms!

Line 77: There are several reviews on plastic ingestion by marine organisms. I suggest using one of them instead of a paper that is specifically on trophic transfer of microplastics.

Line 86: Poorly constructed sentence. Avoid repeating “in the gyre” twice. Also, the microplastics are not all broken down from larger plastics – there are also primary microplastics.

Line 90-91: Highly populated southern hemisphere cities can also be a source, as well as low-income areas that are not heavily populated. Also, the reference (which is incorrect – the paper is from 2014, and missing the “et al.”) is not adequate for this statement.

Line 93: What does distance have to do with island populations contributing litter? Rephrase this.

Line 94: “Most of the islands at this region…”

Lines 97-101: Merge these two sentences to avoid so much repetition.

Line 107: Change “Just as” to

Line 109: Invert the order of this sentence. Start by stating that the South Pacific harbors extensive coral reef ecosystems. I also suggest further exploring the importance of coral reef services to humankind.

Line 114: Change “;” to “,”

Line 115: Explain what are scleractinian corals.

Line 121/122: “…threshold for these corals…”. Do not start both sentences with “If the”. In terms of content, better explain why increasing temperature and acidity affect corals.

Line 125: “…has been tested...”

Line 126: Merge this sentence with the previous phrase. Too much repetition.

Line 128: Start by stating what you are talking about. It seems like you are continuing to talk about scleractinian corals, but you aren’t.

Line 137: change “analyzed” to “evaluated”

Line 138: “ecosystems”. “This work aimed to…”. It is not just a project anymore.

Line 141: “…South Pacific Island of…”. Remove the “to” after the number 2)

Line 142: “…water column of the fringing…”

Line 143: Change the second “determine” to another verb.

Line 151: “…that encircles…”

Line 157: “Additionally, we sampled at two other collection sites:…”

Methods
Provide more details in this section. Throughout methods: do not repeat Richard B. Gump Research Station, in Cook’s Bay, Mo’orea, French Polynesia. State this once and then just refer to it as the Gump Research Station.

Why was the plastic not characterized according to the UNEP guide? What sizes and types of plastics were sampled? This needs to be done and stated.

Line 166: How many sites?

Line 173: Remove parenthesis.

Lines 175-177: Merge these two sentences.

Line 182: How was this pick up conducted? Randomly, considering only larger items? Provide more details here.

Line 184: What does this mean? Percent plastic and plastic?

Line 188: By Km? Do you mean the closest center, as determined by distance in km?

Line 191: Change to “Microplastics study”

Line 195: What is the exact mesh size? Were these tows at the surface, or deeper? Provide more details on your sampling method.

Line 195: Remove sentence starting at: “The net was designed to filter out water…”. This does not need to be stated, as it is the purpose of a net.

Line 196: Provide exact location of net tows in map.

Line 198-200: These sentences should be after you finish describing the tows (i.e. amount of water that was surveyed).

Line 202: Change to: “Each plastic was measured and photographed, and the number of plastics was divided by the surveyed area, to provide the concentration of plastics at the area (number of plastics per square meter).

Line 204: Why is this here? At most this should be discussed and not inserted in the methods. Remove from this section.

Line 205: Why would you want to compare only with gyres?

Line 209: Change to “Plastic processing for coral experiment”

Lines 214-218: “Plastic found on the seafloor of Temae was used in the study because of its negative buoyancy: 5 mm pieces of plastic were taken from the sandy substrate to Gump Station, and smashed with a hammer until they matched the size of observable microplastic in the water column (0.2-1mm).”

Line 224: Remove “happened to be”.

Line 225: Remove “Lab study”. Change to “Exposing Corallimorphs to Microplastics” as in the results.

Line 227: Collected how? 1-3 m water in what way? Depth?

Line 232: What were the laboratorial conditions? Temperature, aeration, where did the water come from? If from the ocean, was it filtered before the experiment?

Line 242: Why 108 hours? Would this time be sufficient for the coral to expel the plastic?

Line 247: Unnecessary to start by stating this. Add this to the end of the paragraph: “All statistical testes were conducted in R (R Core Team, 2016).” You do not have to state where figures were constructed.

Line 254: Feeding trial

Line 255: Differences of what?

Results
Change all “p-value” to just “p”. Standardize the spaces between p and >,< or =

Line 265: Change to “Mean plastic amounts varied by site type…”

Line 269: Change to “...but was not significantly lower than residential beaches (p >0.05).”

Line 274: Do not use “In the end”.

Line 276: Rephrase this paragraph, it is very confusing. What are you correlating? Regression equations should be removed from here and added to the figures.

Line 281: Change to “Microplastic concentration”.

Line 283: Change to “Microplastic was found in the water column at the collection site, with a total of four pieces found in the six tows.”

Line 284: Change “clearly” to “assumed to be”

Line 285: The intertidal of where? Specify where you are referring to, or you could be talking about the entire world.

Line 288: Remove “Lab study”

Line 290: Remove “n” from both sample numbers

Line 292: Remove the second instance of “plastic particles”. It is already clear what you are referring to.

Line 292: What are these means? State them here.

Line 295: Remind the reader what the green plastics are. Also, “green plastics” is repeated three times in this sentence.

Line 296: What are the treatments? Time? Make this clear.

Line 298: Change to “Additionally, a yellow plastic particle was found in an organism in the first treatment, presumably present in the tissue before the experiment”.

Line 300: Do not repeat that the yellow particle was found. Also, “wild-type” is definitely not adequate for plastics.

Additionally, where is the comparison of consumption of microbead vs. naturally-encountered plastic? And the comparison of turbinaria vs. non-turbinaria beaches? And what were the percentages of plastics in your samples? Everything that is stated in the methods should have the corresponding result described in the results section.

Discussion

The discussion is quite poor and needs work.

Line 308: Do not start your discussion with “it”. Simply start with “Natural areas, as opposed to…”

Line 313: This citation is incorrect. Reisser et al. sampled in coastal waters and not beaches. Review this sentence and the reference.

Line 316: This should be together with the first paragraph. Also, do not use “however” at the end of the sentence, it should be at the beginning.

Line 321: Remove first comma.

Paragraph starting at 323: unless you can relate the circulation to the actual points where you sampled, this paragraph does not belong in the discussion.

Line 333: This was not highlighted in the results. Did turbinaria affect the amount of plastics?

Line 354: How does this contribute to the statement that microplastics are the most abundant type of pollution? Also, this reference is not correct, as Reisser et al. did not study microplastics.

Line 357: use :, not ;

Line 361: Use the same number of decimal places on your values. Also, these comparisons are only valid if all these studies used the same mesh sizes.

Lines 365 and 370: change the term “surprising” to “unexpected”. Remove the second instance of this word as well.

Line 367: “pretty” is not a formal measure. Avoid such words.

Paragraph starting at 371: I suggest that this paragraph should be dedicated only to waste management. Finding one plastic that could have come from a larger piece is not significant evidence that the plastic pollution comes from coastal sources.

Line 381: Remove the “F” before “The”

Conclusions
Line 407: You cannot affirm this…

Line 410: Burying or burning trash? The two sentences are stating different things.

Line 416: Health of islanders? How do you know this?

Section “Laboratory study”: this whole section needs to be reviewed and re-worded. Highlight your main findings and better discuss them.
Acknowledgements
This section should be formal and only acknowledge the persons and institutions that aided directly in the work. Avoid such informality as in the last sentence.

Figures
Review all figure legends. You do not have to state that you are showing a graph in the beginning of the legends. Standardize how you refer to sites and use this explanation in all relevant legends. They should be self-explanatory.

Figure 1: Provide a geographical context for this map so the reader knows where in the world the study was conducted. Provide exact microplastic and corallimorphs sampling points, and indicate this in the map legend.

Figure 2 is not cited anywhere in the text. It is not an essential figure, and should be, at most, a supplemental file.

Figures 3 and 4 should be one figure only.

Figures 5 and 6 as well.

Figure 7: The yellow should correspond to one item – it seems like more in the bar.

---

## Round 0.2 · Minor Revisions

I have heard back from one reviewer, who has noted the manuscript is much improved, and I agree with this assessment. However, much work is still needed on focusing and streamlining the writing. I recommend finding a more experienced colleague to help you clean up this paper before any resubmission. I look forward to seeing your resubmission.

Reviewer 2 ·

Basic reporting

The article has improved and I definitely look forward to seeing it published, but another substantial review is required to improve general aspects of writing. I suggest that the author seek another colleague with more experience to review the paper once again before resubmitting.

Experimental design

No comment.

Validity of the findings

No comment.

Additional comments

General comments:

The paper has improved and I definitely look forward to seeing it published, but another substantial review is required to improve general aspects of writing, e.g. remove unnecessary words, unnecessarily short sentences (try merging some of them), and repetition. For instance species name is cited five times in abstract. Microplastics are defined three times in abstract (once in abstract and once in main text is enough). Lines 131-133 have “plastic pollution” repeated three times. Also, avoid beginning subsequent sentences with the same word.

Standardize the manner in which subsections are capitalized (i.e. sentence case, title case, etc.).

Specific comments:

Title: I suggest changing back to “Distribution and biological implications…” The title should follow the general logic of the paper – you first state the problems/methods with macro/microplastics found in the environment, then the problem with ingestion by corallimorphs. Therefore, distribution should come first in the title.

Line 51: “wild” plastic is still stated in the abstract.

Line 53: a separate study? This is one study, which should be divided by research questions.

Line 70: not only by waves…

Line 74: CAN bioaccumulate…this is still subject to debate.

Line 81: Eriksen is misspelled. Also, these two sentences should be merged.

Lines 91-94: merge these two sentences.

Line 116: reword to make clearer that they cannot efficiently extract CARBONATE (not carbon) IF CO2 levels increase past a determined point (that is NOT necessarily a “lethal limit”).

Line 123 (and first section of paragraph): see paper by Lin et al. (2016) “Corallimorpharians are not “naked corals”: insights into relationships between Scleractinia and Corallimorpharia from phylogenomic analyses”.

Line 134: Replace “into the 21st century” with “in the future”.

Line 147: change to “populated”.

Line 152: year?

Line 154: what do you mean the microplastic study? This is all one study. Distinguish in terms of study questions.

Line 162: macroplastic is >5mm.

Line 177 onwards: not necessary to repeat so many times that the survey was 5 minutes.

Line 183: other types of common litter should be in results.

Line 200: remove “concentrated seawater from the surface tows”. This is already clear.

Line 204: change to “microplastics”.

Line 212: not necessary to repeat that they were taken from the sandy substrate.

Line 217-218: “in color” can be removed.

Line 226: not necessary to define “benthic”.

Line 230: °C

Line 232: just “Ten organisms were maintained as control” is sufficient.

Line 253: “This” test.

Line 254: “hourly” rates.

Line 261, and line 290: The three short paragraphs of both sections can be grouped into one.

Line 262: similar “amounts” of plastic.

Line 276: remove “plastic pollution”.

Line 278: remove period before parenthesis.

Line 296: “did not” vary.

Line 312: state that this was not significant.

Line 313: “This distribution of plastic on the island is somewhat surprising when compared to the literature, which generally find larger litter loads near urban areas (Garrity and Levings, 1993; Ryan et al. 2009).”

Line 324: “significantly”.

Line 326: algae do not have spines.

Line 335: “Microplastics in waters of Mo’orea”

Line 346: “compared to”.

Line 347: This should be a continuation of the previous paragraph. Also, change to North America and the Caribbean.

Line 351: “…is a relatively densely populated city that may contribute to the plastic on Mo’orea”. Also, add to this discussion based on Reisser et al.’s findings on plastic near Fiji - Marine Plastic Pollution in Waters around Australia: Characteristics, Concentrations, and Pathways.

Line 354: how was this compared when you only had four microplastic pieces?

Line 364: “The high concentration of plastic and low water flow in the experiment were ideal conditions for plastic consumption; future studies more similar to the actual reef environment are necessary for predicting rates of plastic consumption by wild corallimorphs”.

Line 370: remove first “negatively buoyant”.

Line 373: “It has been previously hypothesized that corallimorphs consume zooplankton or absorb dissolved organic material at an increased rate to survive temperature changes (Kuguru, 2007), so it is possible that under the induced stressful conditions (no water flow), the corallimorphs altered their feeding rate and consumed more plastic.”

Line 380: “of” understanding.

Line 381: remove “greater”.

Line 387: there is no conclusion on the ingestion experiment?

Line 389: merge these two or even three sentences. All three start with “plastic”, this should be changed.

Line 400: the acknowledgements are quite informal and should be reworded.

Figure 1: world map is distorted. Legend: “Location and study sites of Mo’orea Island”.

Figure 2 legend: “Number of macroplastic pieces collected in five-minute surveys, according to (A) site type and (B) the presence of T. oranata”.

Figure 3: why is percent plastic in this analysis? This is not stated in the methods or results. I suggest removing this from the figure.

Figure 4: I suggest changing the colors of the bars to the actual colors of the plastics. Not necessary to have coordinates in the legend. Change to “Number of plastic particles consumed by Discosoma nummiforme according to treatment time and plastic color.”

---

## Round 0.3 · Minor Revisions

The manuscript is now almost acceptable for publication; there remain only some minor edits to be done from my own final personal review of the paper. However, some of these edits appear to be careless misses (e.g. references) so please pay due to attention to correcting all mistakes.

Please correct the following:

1. line 266: spelling of "census".
2. line 269: "plastics" should be singular.
3. line 468 and numerous locations after this: There needs to be a space in "T.ornata" between the genus and species name. In other areas it is spelled "oranata" (line 762) - check all and ensure species names are spelled correctly throughout the paper.
4. line 691: "are rapidly changing" sounds like they are changing due to climate change or some phenomenon. Do you mean "change rapidly"?
5. line 738, 861: Check spelling of corallimorphs. Also, this should be singular or verb needs to be altered. Check spelling of this word throughout the text.
6. line 755-758: While you did find a piece of yellow plastic in a corallimorph, you have not demonstrated "impacting corallimorphs" - the effect was not investigated and is not known. Please delete or lessen the tone of this sentence.
7. References are not formatted in one form; capitalization etc is irregular. Please correct all irregularities.
8. Line 691: Hench should be Hench et al. Similarly, Kuguru (line 248) should be Kuguru et al. Please check ALL references again carefully, it seems there are many misses, particularly using "et al.".
9. Line 957: Hoegh-Guldberg et al. reference not in text?

---

## Round 0.4 · accepted · Accept

The manuscript is acceptable, and the revisions I specifically asked for were well done. However, there are some edits remaining, please ensure these are done no later than the proof stage (and earlier if possible). If possible, take the onus of checking more than I have specifically asked for (e.g. check ALL references) upon yourself, this will speed up acceptance for your future papers.

The following references are missing in the text or need checking as there are irregularities.
Anthony 2016 not in text
Derriak 2002 not in text
Eriksen et al. 2014 not in text
Fitt et al. not in text only Fitt
Mumby et al. wrong year?
UNEP and GRID-Arnel as UNEP only?
Veron et al. 2009 not in text

I look forward to seeing the published version of this paper.